# A Geometric Perspective on the Difficulties of Learning GNN-based SAT Solvers

**Geri Skenderi**
Department of Computing Sciences, Bocconi University, Milan, Italy
Bocconi Institute for Data Science and Analytics (BIDSA)
geri.skenderi@unibocconi.it

## Abstract

Graph Neural Networks (GNNs) have gathered increasing interest as learnable solvers of Boolean Satisfiability Problems (SATs), operating on graph representations of logical formulas. However, their performance degrades sharply on harder and more constrained instances, raising questions about architectural limitations. In this paper, we work towards a geometric explanation built upon graph Ricci Curvature (RC). We prove that bipartite graphs derived from random $k$-SAT formulas are inherently negatively curved, and that this curvature decreases with instance difficulty. Given that negative graph RC indicates local connectivity bottlenecks, we argue that GNN solvers are affected by oversquashing, a phenomenon where long-range dependencies become impossible to compress into fixed-length representations. We validate our claims empirically across different SAT benchmarks and confirm that curvature is both a strong indicator of problem complexity and can be used to predict generalization error. Finally, we connect our findings to the design of existing solvers and outline promising directions for future work.

## 1 Introduction

The Boolean Satisfiability Problem (SAT) is a cornerstone problem of theoretical computer science. It can be succinctly described as the problem of proving logical formulas, whether that consists in making a decision (*is the formula satisfiable?*) or providing an assignment (*which is a bitstring that satisfies the formula?*). SAT plays a foundational role in complexity theory as the first NP-Complete problem(Cook, 1971). Furthermore, many real-world tasks such as circuit verification, automated planning, and software testing, are routinely cast into SAT form (Bordeaux et al., 2006; Marques-Silva, 2008; Biere et al., 2009). Beyond computer science, tools from statistical physics have been used to study typical-case theoretical properties (Monasson & Zecchina, 1997; Krzakała et al., 2007; Zdeborová, 2009) and propose various solvers (Braunstein et al., 2005; Marino et al., 2016; Angelini & Ricci-Tersenghi, 2019).

Recently, Graph Neural Networks (GNNs) (Scarselli et al., 2008; Gilmer et al., 2017) have emerged as a new paradigm in combinatorial problem solving by learning over the graph representations of these problems (Cappart et al., 2023). This trend has been extended to learning SAT solvers (Selsam et al., 2019; Ozolins et al., 2022; Freivalds & Kozlovics, 2022; Chang & Liu, 2025). By representing logical formulas as bipartite graphs that connect variables to clauses, GNNs can be trained end-to-end on both decision and assignment scenarios. Despite their flexibility, these neural solvers remain fragile in practice. Their performance deteriorates on problems that are anecdotally or formally known to be of increasing (algorithmic) difficulty, e.g., at increasing $k$ values in random $k$-SAT (Skenderi et al., 2026; Li et al., 2024).

In terms of representation learning, GNNs suffer from two prevalent issues: oversmoothing (Li et al., 2018) and oversquashing (Alon & Yahav, 2021). Oversmoothing refers to the idea that repeated aggregation causes node representations to become similar, eventually making them indistinguishable. This effect often imposes a practical upper bound on GNN depth. Oversquashing occurs when information from an exponentially expanding neighborhood must be compressed into finite-dimensional embeddings. This bottleneck severely restricts the ability of GNNs to model long-range dependencies. Oversquashing can, therefore, be thought of as a vanishing gradient problem. Given that the difficulty

of SAT can actually be intuited as being directly related to the number of long-range dependencies between nodes, we question whether learning GNN-based solvers on these difficult problems is impacted by oversquashing.

Recent geometric analyses have showed that oversquashing is tightly tied to negative Ricci Curvature (RC) of the underlying graph (Topping et al., 2022; Nguyen et al., 2023). These insights invite a fundamental question: Can graph RC serve as a predictive and constructive notion in unraveling limitations of GNN-based SAT solvers? In this work, we address this question by studying the behavior of the graph RC on random $k$-SAT problems represented as bipartite graphs. We show that, with high probability, the edges become more negatively curved as problems get harder, and less negatively curved as they become easier. Finally, we derive an exact expression for the Balanced Forman Curvature (BFC) in the limit of unsolvable problems, from which we make a connection between curvature and oversquashing in GNNs-based SAT solvers, following the theory of Topping et al. (2022). This is, to the best of our knowledge, the first successful attempt at a theoretical characterization of the limitations of GNN-based SAT solvers.

To validate our theory, we perform experiments on random 3- and 4- SAT problems following Skenderi et al. (2026), and also on different datasets from the recent benchmark of Li et al. (2024). Firstly, we observe a phase transition-like phenomenon in random 3-SAT solving probability as a function of the mean and variance of the BFC. We further affirm the aforementioned limitations by rewiring only the testing graphs of the benchmarks at test-time to increase their average BFC, and show that these rewired problems become much easier to solve. Finally, we find that heuristics based on the average BFC of a dataset correlate well with generalization error, unlike the average clause density, which is typically used to characterize the hardness of a single instance. Overall, our findings suggest that GNN-based SAT solvers suffer from two different types of hardness: the hardness of learning representations of inputs with high negative BFC, followed by the well-established algorithmic hardness of SAT. We conclude by relating our findings with modeling and design principles of existing approaches, as well as providing avenues for future research.

## 2 BACKGROUND

This section is only intended to formally introduce the objects studied in the paper and render the material self-contained. We kindly ask the reader to tolerate the occasional whirlwind and abuse of notation, which will be formalized later in Section 3.

### 2.1 RANDOM $k$ BOOLEAN SATISFIABILITY PROBLEM

The random $k$-SAT (assignment) problem, which is the central object of study in this work, is made up of $N$ variables $\{x_i\}_{i=1}^N$ that can take binary values $x_i \in \{0, 1\}$. Using these variables, one constructs $M$ clauses containing a disjunction of $k$ variables or their negations (called literals). For example, a 3-SAT problem would have clauses of the form $(x_i \vee \neg x_j \vee x_h)$. The goal is to assign a value to all literals such that they satisfy the conjuction of all clauses, a logical formula posed in Conjunctive Normal Form (CNF). In the random formulation, it is possible to identify different phases of problem hardness based on the clause density parameter $\alpha = M/N$. This phenomenon has been actively researched in statistical physics due to the analogies between random $k$-SAT and spin-glass models (Mézard et al., 1987; Mezard & Montanari, 2009). Notable results(Mézard et al., 2002; Krzakała et al., 2007; Zdeborová, 2009) include the discovery of two transitions for typical instances at a given $k$, based on the value of $\alpha$ in the thermodynamic limit ($M, N \to \infty$): As $\alpha$ increases, the measure over the space of possible solutions first decomposes into an exponential number of clusters at the dynamical transition $\alpha_d(k)$ and subsequently condensates over the largest such states at the critical transition $\alpha_c(k)$. These phase transitions naturally affect the performances of many algorithms, e.g., when $k \geq 4$, solving beyond $\alpha_c$ is almost impossible with existing algorithms.

### 2.2 GRAPH NEURAL NETWORKS

GNNs are a subclass of Neural Networks (NNs) that learn representations of graph data by locally aggregating information(Scarselli et al., 2008; Gilmer et al., 2017). The main goal of the architecture is to implement symmetries natural to graphs (Bronstein et al., 2017). Consider, for simplicity, an unweighted and undirected graph $G$ with $N$ nodes, represented by a symmetric binary adjacency

matrix $A \in \{0, 1\}^{N \times N}$. This setting can be easily extended to deal with more general connectivity structures (Corso et al., 2024). By associating a node signal $X \in \mathbb{R}^{N \times m}$ to the graph, we can describe a GNNs as a learnable convolution of the node signal with $A$ as the shift operator. A generalization of this concept can be obtained by considering the message-passing framework:

$$x_i^{(k)} = \theta^{(k)} \left( x_i^{(k-1)}, \bigoplus_{j \in S_1(i)} \phi^{(k)} \left( x_i^{(k-1)}, x_j^{(k-1)}, e_{ji} \right) \right), \tag{1}$$

where $x_i^{(k)}$ denotes node features of node $x_i$ at layer $k$, $e_{ji}$ the (optional) edge features from node $j$ to node $i$, $S_1(\cdot)$ the set of (1-hop) neighbor nodes, $\bigoplus$ a differentiable, permutation invariant function, (e.g., sum, mean), and $\phi, \theta$ denote differentiable and (optionally) nonlinear functions such as Multi-Layer Perceptrons (MLPs).

A CNF formula can be readily translated into a bipartite graph (Biere et al., 2009), which can then be fed into a GNN-based solver. The particular bipartition we consider in this work is detailed later in Section 3. The application of the above message-passing scheme on SAT problems can be seen as using Equation1 for the clause and literal partitions (Li et al., 2024). Let $i$ be a literal node and $j$ be a clause node, then:

$$\begin{aligned}
x_j^{(k)} &= \theta_c^{(k)} \left( x_j^{(k-1)}, \bigoplus_{i \in S_1(j)} \phi_l^{(k)} \left( x_i^{(k-1)} \right) \right), \\
x_i^{(k)} &= \theta_l^{(k)} \left( x_i^{(k-1)}, \bigoplus_{j \in S_1(i)} \phi_c^{(k)} \left( x_j^{(k-1)} \right) \right),
\end{aligned} \tag{2}$$

where the subscripts $c$ and $l$ refer to parameters specific to the clause and literal partitions respectively.

### 2.3 RICCI CURVATURE OF GRAPHS

In Riemannian geometry, RC quantifies the local deviation of a manifold $\mathcal{M}$ from flat Euclidean space based on (loosely speaking) volume change. Intuitively, RC captures how the neighborhoods of two adjacent points relate when moving from one point to the other by comparing how a small ball of mass around a point is distorted when transported along a geodesic to a neighboring point. Extending this notion to more general structures, such as metric spaces or combinatorial complexes, is an extremely active area of mathematical research, with the works of Ollivier (2009) and Forman (2003) standing out. In the case of graphs, we ask ourselves how local connectivity either concentrates or disperses. Ollivier (2009) implements this idea by comparing probability mass on local neighborhoods, i.e., a random walk distribution on the endpoints of an edge. Given these two distributions, one can compare the ratio between their Wasserstein and shortest path distance, serving as a direct and discrete analogue of geodesic transport. See Appendix A.1.1 for more details. The definition of Forman (2003) relies heavily on topology, and thus it takes a combinatorial form. Essentially, given a cell complex, the curvature of a p-cell depends only on the topological structures between the cell and its neighbors. This makes it quite simple to compute numerically. Given that RC is directly related to the structure of local neighborhoods, it has emerged as a powerful way of theoretically analyzing limitations of GNNs. In a seminal paper, Topping et al. (2022) provide both a balanced version of the Forman-Ricci Curvature (FC) curvature and show that the oversquashing problem (Alon & Yahav, 2021) can be directly connected to edges with high negative curvature. This definition, namely the BFC, is central to this paper, therefore please consult Appendix A.1.2 for the definition and additional details. Nguyen et al. (2023) have shown that similar results can be derived using the Ollivier-Ricci Curvature (OC) curvature. It is worth noting that these notions of curvature are naturally correlated with one another, as shown empirically in a multitude of complex networks by Samal et al. (2018).

## 3 CURVATURE OF RANDOM k-SAT AND ITS RELATIONSHIP WITH GNNS

**Setting and Notation.** We consider random $k$-SAT problems with $N$ variables and $M$ clauses, with $\alpha = M/N$ and $k, N, M \in \mathbb{N}$. The problems are represented as simple bipartite graphs $G = (V, E)$, where the node set is a literal-clause bipartition $V = L \cup C$, with $L \cap C = \emptyset$ and $|L| = 2N, |C| = M$. The edge set takes the form $E = \{(i, j) \in V \times V : i \sim j, i \in L, j \in C\}$. Given $v \in V$, we

denote its degree by $d_v$. The expected value of the random variable $X$ with probability distribution $P$ is denoted by $\mathbb{E}[X]$, while $\mathbb{E}_{p \sim P}[X]$ is expectation over samples drawn from $P$. Unless noted otherwise, when we refer to the expected value in simulations, we imply its estimate via the sample mean statistic. We denote by $X_n \xrightarrow{\text{P}} c$ the convergence in probability of a sequence of random variables to a constant $c$ under a given limiting behavior Finally, we denote the BFC at an edge $i \sim j$ by $\kappa(i, j)$. We remind the reader that we are going to analyze this variant of graph RC, defined in Appendix A.1.2.

**Data Model.** Our bipartite formulation is a simplification of the input graphs considered in many GNN-based solvers (Selsam et al., 2019; Ozolins et al., 2022). Recent literature (Li et al., 2024) refers to this data structure as a Literal-Clause Graph (LCG). We consider an Erdős–Rényi-like procedure, where each clause is assigned $k$ distinct literals independently at random with probability $p$. Assuming therefore that all literals are equally likely to appear in a given clause, we obtain the following degree distributions:

$$P(d_j = h) = \delta(h - k), \quad P(d_i = h) = \binom{M}{h} p^h (1-p)^{M-h}, \tag{3}$$

where $\delta(\cdot)$ represents the delta function, $\binom{\cdot}{\cdot}$ the binomial coefficient, and $p := \frac{k}{2N}$. In the limit $M, N \to \infty$, we can approximate the Binomial form of the literal degree distribution with a Poisson distribution:

$$P(d_i = h) = e^{-\lambda} \frac{\lambda^h}{h!}, \tag{4}$$

with $\lambda = Mp = \frac{1}{2}\alpha k$. We will be interested in using this approximation to analyze the graph BFC, which is an edge level property that is dependent on the degrees of the endpoints. Therefore we need to consider sampling on the edges of the graph. This fact has no implications on $d_j$ given its distribution, but it does imply that for our calculations, the literal degree should have zero mass on $d_i = 0$ and be biased towards higher-degree literals under edge sampling. This line of reasoning naturally leads to the idea of considering a size-biased PMF (Shanker et al., 2021) as follows:

$$P^*(d_i = h) = \frac{hP(d_i = h)}{\mathbb{E}[d_i]} = e^{-\lambda} \frac{\lambda^{(h-1)}}{(h-1)!}, \tag{5}$$

which can easily be seen to be an equivalence in distribution as $P^*(d_i = h) = P(d_i = h-1), \forall h \geq 1$.

**Characterizing curvature in easy and hard regimes.** We will now proceed by showing that the BFC of the above bipartite representation can be characterized in probability. This, in turn, will have implications for oversquashing. The proofs of all statements are deferred to Appendix A.2.

**Proposition 3.1** (BFC bounds on bipartite graphs). Define the quantity

$$\underline{\kappa}(i, j) = \begin{cases} 0 & \text{if } \min\{d_i, d_j\} = 1 \\ \frac{2}{d_i} + \frac{2}{d_j} - 2 & \text{otherwise} \end{cases} \tag{6}$$

For any edge $i \sim j$ in a simple, unweighted bipartite graph $G$, we have that that $-2 < \underline{\kappa}(i, j) \leq \kappa(i, j) \leq 1$.

**Proposition 3.2.** Let $i \sim j$ be an edge from the LCG representation $G$ of a random k-SAT problem with $N$ variables and $M$ clauses, with degree distributions given by Equations 5 and 3. Then $\underline{\kappa}(i, j) \xrightarrow{\text{P}} 0$ as $\alpha = \frac{M}{N} \to 0$.

Proposition 3.2 tells us that as problems become easier, their graph representations get (Ricci) flatter. This implies a relationship between trivial problems and their bipartite topology, which is that each clause is made of distinct, unique literals. In this scenario, producing a satisfying assignment becomes easy since assigning truth values to literals of one clause does not affect other clauses. This implication aligns well with common knowledge, therefore the next thing to check is the behavior when dealing with very difficult problems. This later case is important because it is close to the unsatisfiable phase where the faults of existing algorithms emerge (Angelini & Ricci-Tersenghi, 2019). We can formalize this intuition by first looking at the behavior of the lower bound in the limit of unsatisfiable when problems.

**Lemma 3.1.** Let $i \sim j$ be an edge from the LCG representation $G$ of a random k-SAT problem with $N$ variables and $M$ clauses, with degree distributions given by Equations 5 and 3. Then $\underline{\kappa}(i,j) \xrightarrow{\text{P}} \frac{2}{k} - 2$ as $\alpha = \frac{M}{N} \to \infty$.

In the unsatisfiable limit, the lower bound of the graph BFC will tend to be maximally negative with high probability. Furthermore, Lemma 3.1 provides an interesting interplay between $\alpha$ and $k$ in the aforementioned limit. We now show that this behavior holds in fact also for the BFC, and explore all the implications.

**Theorem 3.1.** Let $i \sim j$ be an edge from the LCG representation $G$ of a random k-SAT problem with $N$ variables and $M$ clauses, with degree distributions given by Equations 5 and 3. Let $\sharp_\square^i(i,j) := \{n \in S_1(i) \setminus S_1(j), n \neq j : \exists w \in (S_1(n) \cap S_1(j)) \setminus S_1(i)\}$ be the neighbors of $i$ forming a 4-cycle based at $i \sim j$, *without* diagonals inside. The BFC at $i \sim j$ is bounded from above by the quantity:

$$\bar{\kappa}(i,j) := \frac{2}{d_i} + \frac{2}{d_j} - 2 + \frac{|\sharp_\square^i(i,j)| + |\sharp_\square^j(i,j)|}{d_i}. \tag{7}$$

Furthermore, in the limit $N \to \infty$, $k$ fixed, $M = \alpha N$, and $\alpha \to \infty$, $\kappa(i,j) \xrightarrow{\text{P}} \frac{2}{k} - 2$.

Theorem 3.1 contains a crucial result in its probabilistic characterization of the BFC, which is that it is connected to both $\alpha$ and $k$ in the same way as shown for $\underline{\kappa}(i,j)$ above. As the problems get harder, the number of constraints becomes a decisive factor on the curvature. A larger value of $k$ implies more constraints and thus many long range interactions between the literals, and it also implies that edges tend to become more negatively curved. This phenomenon can be visually observed in Figure 3 (Appendix), where for smaller $k$, larger values of $\alpha$ are required to have highly negatively curved edges. This latter point is crucial in developing a more profound understanding of the limitations of GNN-based solvers, and we will expand upon it in the following discussion.

**Message-Passing Bottlenecks and Downstream Performance.**   The results presented in the previous section allow us to proceed with a principled way of understanding an important limitation for GNN-based SAT solvers. This is due to the direct connection between our result in Theorem 3.1 with Theorem 4 of Topping et al. (2022), which establishes that "edges with high negative curvature are those causing the graph bottleneck and thus leading to the oversquashing phenomenon". This seminal result states that if the gradients of the message passing functions ($\theta$ and $\phi$ in Equation 1) are bounded, and there exists a sufficiently negatively curved edge compared to the degrees of its endpoints, then the derivative of the learned node representations around that edge vanishes. Intuitively, this can be understood as a difficulty of propagating the information in nodes at a reachable distance due to the fact that the graph structure limits the pathways where information can travel: nodes in different neighborhoods need to pass all messages through the same edge. Formally, for larger values of $k$, as the clause density $\alpha \to \infty$, any infinitesimally small value $\delta > 0$ could be used in Theorem 4 of Topping et al. (2022) such that $\kappa(i,j) \leq -2 + \delta$, leading to an exponentially decaying Jacobian of the node representations around $i \sim j$. This result leads us to the conclusion that *GNN-based solvers are limited from two distinct hardness types: the algorithmic hardness inherent to SAT and the hardness of learning representations for long range communication.* The interplay between $k$ and $\alpha$ in Theorem 3.1 provides additional insights. Indeed, for problems with large values of $k$ or large values of $\alpha$, highly negatively curved edges are guaranteed to exist on average, and this quantity will concentrate. On the other hand, for large values of $\alpha$ and relatively small values of $k$, the latter becomes crucial in deciding how well a GNN-based solver will be able to learn, i.e., it should be easier to learn a solver for smaller values of $k$. We confirm this fact empirically in Section 4.

The intuition we provide for a more complete understanding of the discussion above is the following: At increasing connectivity, literals become very distant on the interaction network, i.e., the number of long-range codependencies increases. In this scenario, a GNN will not be able to learn a fixed length representation that can "remember" the information of reachable, but not directly adjacent nodes. This means that the ability to learn a solver is compromised by an oversquashing phenomenon. For large values of $k$, this problem becomes prevalent even before the hardness of exploring the solution space, due to the effect of $k$ on the BFC. Our theory motivates therefore how increasing values of $k$ in random $k$-SAT would lead to worse oversquashing and performance, even for what would be considered simple problems in terms of $\alpha$. To visually see this, let us consult the level curves in Figure 3 (Appendix). As the value of $k$ grows, the gap between the flatter (yellow) and

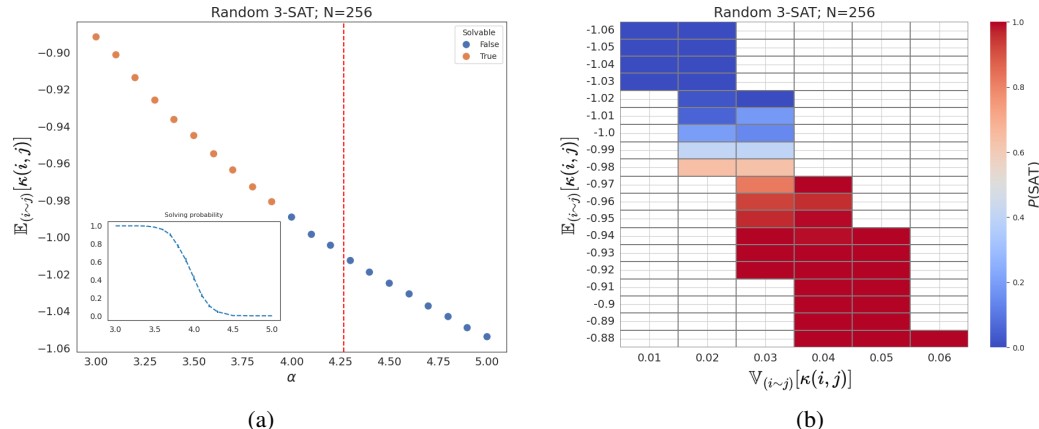

Figure 1: *(a)* Average BFC as a function of $\alpha$ for random 3-SAT problems with $N = 256$. The color is used as a representation for the average solvability of the problems at a given $\alpha$ by the NeuroSAT model (Selsam et al., 2019), with a group labeled as solvable if 50% or more of the problems get a satisfying assignment. The vertical red line represents the analytical SAT-UNSAT critical threshold $\alpha_c \approx 4.267$ (Mertens et al., 2006). The average BFC drops monotonically with $\alpha$. The small plot in the bottom-left corner provides the model's solution probability curve in terms of $\alpha$, where it is possible to notice the algorithmic transition from satisfiable to unsatisfiable problems. *(b)* Probability of finding a satisfying assignment of the same problems as in (a) with NeuroSAT as a function of the variance and mean of the BFC. Notice how as $\alpha$ grows, the average curvature not only gets more negative, but also concentrates. We can see from the empirical results in (a) that in this case, the model is unable to produce a satisying assignment. As $\alpha$ becomes smaller, the input graphs have less negative edges on average, the associated variance naturally grows, and so does the solving probability. Using the first two moments of the BFC, we are able to observe a similar transition-like phenomenon as the small plot in the bottom-left corner of (a). Best viewed in color.

highly negatively curved problems (violet) gets smaller. The same holds for increasing values of $\alpha$, as expected. We provide more visual depictions of the additional implications of this phenomenon on the long-range codependencies in Appendix A.5, where we plot two input graphs for random 3-SAT at small (Figure 4) and large (Figure 5) $\alpha$. In the following section, we will empirically confirm our results for two different GNN-based solvers.

## 4    EXPERIMENTS

**Experimental Setting.**   To validate our theory, we perform different experiments, the details of which will be provided in the respective subsections. We first explore the behavior of GNN-based solvers as a function of the input graph BFC. Based on these results, we then propose two heuristics to understand how hard a given SAT dataset will be to solve for a GNN-based solver. We focus on the assignment scenario, as it includes the decision scenario as well. Given that all the datasets we utilize do not come with node features, we use learnable embeddings in order to effectively explore oversquashing. The GNN-based solvers are implemented following the design of Li et al. (2024), using PyTorch (Paszke et al., 2019), PyTorch-Geometric (Fey & Lenssen, 2019) and PyTorch Lightning (Falcon & The PyTorch Lightning team, 2019). The networks were trained for 100 epochs using the AdamW optimizer (Loshchilov & Hutter, 2019), with learning rate $\eta = 0.0001$ decaying by half after 50 epochs and the gradients clipped at unit norm, on NVIDIA Titan RTX GPUs.

### 4.1    THE RELATIONSHIP BETWEEN CURVATURE AND SATISFIABILITY

We start by using numerical simulations to verify the theoretical claims made in Section 3. To do so, we utilize the $k-$SAT benchmarks of Skenderi et al. (2026), which consist of random instances in Conjunctive Normal Form (CNF) implemented in the Julia language (Bezanson et al., 2017). The problems are generated with $\alpha \in [3, 5]$ and $N \in [16, 32, 64, 128, 256]$, using steps of $\Delta\alpha = 0.1$,

Table 1: Accuracy of producing satisfying assignments on SAT benchmark datasets (Li et al., 2024) of two different GNN-based solvers. By increasing the testing set's curvature at test-time through rewiring, both solvers are able to make big leaps in accuracy, especially in more constrained problems. Reducing the curvature of the problem facilitates long range communication and renders problems easier. The reported results represent the average and standard deviation over 5 different runs, with the **best results** (for each model) in **boldface** and absolute improvements in parentheses.

| Model | Variation | Datasets | | | |
|---|---|---|---|---|---|
| | | 3-SAT | 4-SAT | SR | CA |
| GCN | No Rewiring | $0.510 \pm 0.012$ | $0.180 \pm 0.048$ | $0.470 \pm 0.031$ | $0.650 \pm 0.016$ |
| | *Test-time Rewiring* | ***0.626 ± 0.021** (+0.116)* | ***0.374 ± 0.045** (+0.194)* | ***0.696 ± 0.035** (+0.226)* | ***0.670 ± 0.048** (+0.020)* |
| NeuroSAT | No Rewiring | $0.690 \pm 0.022$ | $0.436 \pm 0.032$ | $0.734 \pm 0.017$ | $0.746 \pm 0.018$ |
| | *Test-time Rewiring* | ***0.820 ± 0.030** (+0.130)* | ***0.686 ± 0.029** (+0.250)* | ***0.902 ± 0.004** (+0.168)* | ***0.828 ± 0.029** (+0.082)* |

thereby capturing both satisfiable and unsatisfiable regimes around the critical threshold $\alpha_c$ (Mertens et al., 2006). Considering its widespread use and downstream performance, we train the NeuroSAT model (Selsam et al., 2019) to produce a satisfying assignment, while scaling the number of message passing iterations by $2N$ during evaluation, which has been shown to a good scaling regime to balance efficiency and accuracy (Skenderi et al., 2026). We then analyze the performance of the model on problems with $N = 256$ (for a better statistical correspondence to the theoretical setting), with the results being summarized in Figure 1. Our results show that by considering the probability of finding a solution at a given $\alpha$ as a function of the first and second moments of the curvature, we can replicate a SAT/UNSAT phase-transition (Figure 1b). This result presents an important step forward in theoretically understanding the performance of GNN-based solvers. Similar results hold for random 4-SAT, and can be seen in Figures 6 and 7 in the Appendix. For this higher value of $k$, we have more negatively curved edges which strongly impact the performance of GNN-based solvers, as will further confirmed in Section 4.2. An interesting observation is that for random 3-SAT, the curvature starts to become highly negative and concentrate close to the dynamical threshold $\alpha_d \approx 3.927$ (Mertens et al., 2006).

**Test-time Rewiring.** In order to obtain additional evidence about the previous observations, we put ourselves in a unique scenario: Suppose a GNN-based solver is trained on a dataset of SAT problems, and later tested on a separate testing partition. If we render the said testing partition less curved, would the model perform better without needing to retrain? The purpose behind this experiment is to gain a deeper understanding of the relationship between curvature and problem complexity. For this purpose, we use four different SAT benchmarks proposed by Li et al. (2024): Random 3 and 4 -SAT generated only near their (respective) critical thresholds $\alpha_c$, a random $k$-SAT dataset consisting of mixed $k$ values (SR), and one that mimics the modularity and community structure of industrial problems (CA). The last two datasets are better representatives of real-world problems. We train both a Graph Convolutional Network (GCN) solver (Kipf, 2016) and NeuroSAT on training partitions using the same protocol as before, while the testing partition is rewired using a stochastic discrete BFC flow (Topping et al., 2022). The idea behind the rewiring procedure is quite simple: we make the input graph less curved by stochastically deleting edges that have the highest negative curvature, while adding new edges that are less curved. We provide a more detailed explanation of this process, including a schematic algorithm in Appendix A.3. The results are reported in Table 1, where it can be observed that that the rewired problems become simpler to solve for both solvers at test-time. A noteworthy observation is that a large improvement happens on 4-SAT problems, while the modular CA dataset reports small improvements. Albeit already intuitive, we make a direct connection of this result with our theory in the following subsection.

## 4.2 A New Hardness Heuristic for GNN-Based Solvers

Based on the developed theory and the above observations, we provide, as a practical contribution, two different heuristics that reflect how hard it will be for a GNN-based solver to tackle a dataset. The main motivation behind these heuristics is that if we simply look at the average clause density of a dataset, we can miss out on direct implications of oversquashing. An example of this is the first dataset we presented for the random 3-SAT experiments discussed in Figure 1, which is build at

increasing values of $\alpha$. Given an input graph $G$, we define the heuristics as:

$$\omega(G) = -\mathbb{E}_{(i \sim j)}[\kappa(i, j)] \cdot \mathbb{E}[\alpha], \quad \omega^*(G) = \frac{\omega(G)}{\mathbb{V}_{(i \sim j)}[\kappa(i, j)]}, \tag{8}$$

with the expectations being taken over the edges $G$. The averages of both heuristics, which we denote by $\bar{\omega}$ and $\bar{\omega}^*$, can then be used to judge the hardness of a given dataset. Intuitively, we provide two non-negative numbers that reveal how dense and curved $G$ is on average ($\omega(g)$) and how much this quantity concentrates ($\omega^*(G)$). Our theoretical insights tell us that these quantities should provide information into the hardness of learning a GNN-based solver. We report the generalization error (1 - testing accuracy) of NeuroSAT on the four previously mentioned benchmarks in Section 4.1, alongside the heuristics in Table 2. A (linear) correlation analysis between the error and the (normalized) heuristics reveals that our curvature-based approach serves as a better predictor of generalization: the respective correlation coefficients are $\rho_{\bar{\alpha}} = 0.32$, $\rho_{\bar{\omega}} = 0.86$ and $\rho_{\bar{\omega}^*} = \mathbf{0.98}$.

These results allow us to formally motivate the performance gains during the test-time rewiring procedure discussed previously. What we observe, is that due to its community structure, the CA dataset has a large clause density, but its average curvature is much lower than that of random 4-SAT problems. This is natural, since a community structure is inherently linked with edges that act as less important bottlenecks for message passing (Nguyen et al., 2023).

Table 2: Average generalization error and hardness heuristics.

| Problem | Generalization Error | Hardness Heuristic | | |
|---------|---------------------|:---:|:---:|:---:|
| | | $\bar{\alpha}$ | $\bar{\omega}$ | $\bar{\omega}^*$ |
| 3-SAT | 0.31 | 4.59 | 4.12 | 97.41 |
| 4-SAT | 0.56 | 9.08 | 9.81 | 612.32 |
| SR | 0.27 | 6.09 | 5.30 | 125.30 |
| CA | 0.25 | 9.73 | 6.30 | 123.27 |

These results show that the ability of GNN-based solvers to learn representations for long range communication and generalize well is deeply connected with the curvature of the input data.

## 5 CONCLUSIONS

**Practical Takeaways and Future Work.** In this paper, we have identified that the geometry of the input data is a plausible cause of deficiency in the performance of GNN-based SAT solvers, making connections to the oversquashing phenomenon. Tight relationships between data structure and learning are universally prevalent across all applications and as a result we have different modeling principles for different data. Our main takeaway in this work is that a general purpose GNN architecture must be specialized in order to deal with SAT-specific issues, i.e., we cannot hope to learn faithful combinatorial solvers naively. What is quite fascinating is that most modern GNN-based SAT solvers implement some type of recurrence mechanism (Selsam et al., 2019; Ozolins et al., 2022; Li et al., 2024; Mojžíšek et al., 2025), and this architectural component has been recently shown to be a great starting point to mitigate oversquashing (Arroyo et al., 2025). The effect of recurrence can be immediately noted by comparing the drop in performance between the GCN and NeuroSAT solvers in Table 1. A direct avenue for future work that we consider promising in this direction is the application of continuous graph diffusion dynamics for learning(Chamberlain et al., 2021; Han et al., 2024), which can generalize the recurrence mechanism.

This above reflection leads us to conjecture that the relationship between input data and model performance is prevalent throughout Neural Combinatorial Optimization (NCO) (Wang et al., 2022), and different architectural designs are necessary for different problems. Furthermore, while we were able to identify a cause for the hardness of learning GNN-based SAT solvers and also provided heuristics to identify the hardness of a given dataset, we did not propose new modeling principles. One might intuitively suppose that using curvature-aware solvers (Ye et al., 2019; Fesser & Weber, 2024) might provide advantages, but the concentration of the BFC would not make this conjecture plausible. We indeed verify this problem in Appendix A.4, where it can be seen that simple curvature-aware GNNs do not report consistent improvements. How to best inject curvature information in GNN-based solvers remains an open avenue for future work.

**Closing Remarks.** In conclusion, our results highlight that the limitations of GNN-based SAT solvers cannot be fully understood without considering the geometric properties of the input. Our study presents, to the best of our knowledge, the first attempt at a theoretical understanding of

these neural solvers, by establishing a direct connection between their negatively curved graph representations and oversquashing in GNNs. We provide empirical evidence of this connection and verify that it is prevalent on more constrained instances. Beyond SAT, we expect these insights to be valuable for other domains where graph representations of combinatorial problems are employed. Combinatorial Optimization (CO) provides an interesting venue to study the reasoning behavior of GNNs, and we hope that this paper makes a case for such studies. In conclusion, we hope that bridging concepts from deep learning, geometry, and physics, will pave the way for principled advances in the design of neural solvers.

**Acknowledgements.** The author would like to thank the funding project Next Generation EU - MIUR PRIN PNRR 2022 Grant P20229PBZR provided by the European Union and MIUR. Furthermore, many thanks are due to several people that were kind enough to discuss the ideas of this work with the author, namely Dr. Sebastiano Ariosto, Dr. Enrico Ventura, and Prof. Carlo Lucibello. Finally, the views and opinions expressed are those of the author only and do not necessarily reflect those of the European Union or the European Research Council Executive Agency. Neither the European Union nor the granting authority can be held responsible.

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

# A  APPENDIX

In what follows, we provide supplementary material that complements the main manuscript. The appendix is organized as follows:

1. Appendix A.1 begins with background on graph Ricci Curvature (RC), where we review the Ollivier-Ricci Curvature (OC), Forman-Ricci Curvature (FC), and Balanced Forman Curvature (BFC) discretizations to give the reader an intuition for what graph RC describes.

2. In Appendix A.2, we provide the proofs of the formal statements regarding the characterization of the BFC in the easy and hard regimes of random $k$-SAT problems.

3. Appendix A.3 describes the test-time graph rewiring procedure used in our experiments, including a full presentation of the stochastic curvature-guided algorithm.

4. In Appendix A.4, we outline preliminary ideas and implementations for curvature-aware solvers, along with empirical results that illustrate their shortcomings following the discussion in the main paper.

5. Finally, Appendix A.5 contains additional plots and visualizations that further augment the previously presented material.

## A.1  RICCI CURVATURE OF GRAPHS

In this section, we provide, for the sake of completeness, definitions and intuitive explanations of the two types of graph RC mentioned in this paper. The definitions are taken from Bhattacharya & Mukherjee (2015) and Topping et al. (2022) respectively, we simply report them here in a synthesized manner. We kindly refer the reader to the aforementioned works for more details.

### A.1.1  OLLIVIER RICCI CURVATURE (OC)

The formulation of Ollivier (2009) on graphs tries to mimic the intuition from differential geometry: we use a ratio between the amount of "mass" moved around of an edge neighborhood with the shortest path distance, i.e., the graph geodesic. We can see therefore that it is an edge-level, scalar-valued function. Edges with negative curvature act as structural bottlenecks, separating dense regions and limiting smooth information propagation. Edges with positive curvature in the other hand facilitate smooth information propagation and are indicators of community structure. Graph OC is very well studied and has being linked to properties of graph Laplacians and mixing times of Markov Chain Monte Carlo (MCMC) methods (Paulin, 2016; Inagaki, 2025). Let us formalize this concept.

For two probability measures $\mu_1, \mu_2$ on a metric space $(X, d)$, the the *Wasserstein distance* between them is defined as

$$W_1(\mu_1, \mu_2) = \inf_{\nu \in M(\mu_1, \mu_2)} \int_{X \times X} d(x, y) \mathrm{d}\nu(x, y), \tag{9}$$

where $M(\mu_1, \mu_2)$ is the collection of probability measures on $X \times X$ with marginals $\mu_1$ and $\mu_2$. The Wasserstein distance is the result of the solution to a famous problem called Optimal Transport (Peyré et al., 2019). Intuitively, this distance measures the optimal cost to move one pile of sand to another one with the same mass.

Let a metric measure space $(X, d, m)$ be a metric space $(X, d)$, with a collection of probability measures $m = \{m_x : x \in X\}$ indexed by the points of $X$. The (coarse) Ricci curvature of a metric measure space is defined as follows:

**Definition A.1** (Ollivier-Ricci Curvature (Ollivier, 2009))**.** Given the metric measure space $(X, d, m)$, for any two distinct points $x, y \in X$, the (coarse) *Ricci curvature* of $(X, d, m)$ of $(x, y)$ is defined as:

$$\kappa_{OC}(x, y) := 1 - \frac{W_1(m_x, m_y)}{d(x, y)} \tag{10}$$

Extending this definition to graphs requires some additional steps. Consider a locally finite and possibly weighted simple graph $G = (V, E)$, where each edge $i \sim j \in E$ is assigned a positive weight $w_{ij} = w_{ji}$. The graph is equipped with the standard shortest path graph distance $d_G$, that is, for $i, j \in V$, $d_G(i, j)$ is the length of the shortest path in $G$ connecting nodes $i$ and $j$. For $i \in V$

define the degree $d_i := \sum_{i \sim j \in E} w_{ij}$ and the neighborhood $\mathcal{N}(i) := \{j \in V : i \sim j \in E\}$. For each $i \in V$ define a probability measure

$$
m_i(j) = \begin{cases} \frac{w_{ij}}{d_i}, & \text{if } j \in \mathcal{N}(i) \\ 0, & \text{otherwise.} \end{cases}
$$

Note that these are just the transition probabilities of a weighted random walk on the vertices of $G$. If $m_G = \{m_i : i \in V\}$, then considering the metric measure space $\mathcal{M}_G := (V, d_G, m_G)$, we can define the OC curvature for any edge $i \sim j \in E$ as following Equation A.1 and discretizing Equation 9 on $\mathcal{M}_G$:

$$
\kappa_{OC}(i,j) := 1 - W_1^G(m_i, m_j), \quad W_1^G(m_i, m_j) = \inf_{\nu \in \mathcal{A}} \sum_{z_1 \in \mathcal{N}(i)} \sum_{z_2 \in \mathcal{N}(j)} \nu(z_1, z_2) d(z_1, z_2). \quad (11)
$$

$\mathcal{A}$ denotes the set of all $d_i \times d_j$ matrices with entries indexed by $\mathcal{N}(i) \times \mathcal{N}(j)$ such that $\nu(i', j') \geq 0$, $\sum_{z \in \mathcal{N}(j)} \nu(i', z) = \frac{w_{ii'}}{d_i}$, and $\sum_{z \in \mathcal{N}(i)} \nu(z, j') = \frac{w_{jj'}}{d_j}$, for all $i' \in \mathcal{N}(i)$ and $j' \in \mathcal{N}(j)$. For a matrix $\nu \in \mathcal{A}$, $\nu(i', j')$ represents the mass moving from $i' \in \mathcal{N}(i)$ to $j' \in \mathcal{N}(j)$. For this reason, the matrix $\nu$ is often called the *transfer plan*.

### A.1.2  BALANCED FORMAN CURVATURE (BFC)

The FC is a discretization of Ricci curvature that holds for a broad class of topological objects, namely so called (regular) cellular (CW) complexes (Samal et al., 2018). The original definition (Forman, 2003) is both extremely technical and out of the scope of this paper. Given that graphs edges are 1-dimensional cells (topologically), the definition simplifies greatly making use of very simple graph properties. Consider a simple, unweighted graph for simplicity, then the FC of edge $(i, j)$ is defined as $\kappa_{FC}(i,j) = 4 - d_i - d_j$. The main issue of this definition is that it ignores higher order correlations in terms of triangles and 4-cycles, which prove crucial to discriminate positively, flat, and negatively curved graphs. Inspired by these issues, Topping et al. (2022) propose an extension of the FC dubbed BFC, such that it is both fast to compute and encodes accurate curvature information.

**Definition A.2** (Topping et al. (2022)). For any edge $i \sim j$ in a simple, unweighted graph $G = (V, E)$ with adjacency matrix $A$ let:

- $S_1(i) := \{j \in V : i \sim j \in E\}$ be the set of 1-hop neighbors of $i$.

- $\sharp_\Delta(i,j) := S_1(i) \cap S_1(j)$ be the set of triangles based at $i \sim j$.

- $\sharp_\square^i(i,j) := \{k \in S_1(i) \setminus S_1(j), k \neq j : \exists w \in (S_1(k) \cap S_1(j)) \setminus S_1(i)\}$ be the neighbors of $i$ forming a 4-cycle based at $i \sim j$ *without* diagonals inside.

- $\gamma_{\max}(i,j) := \max \left\{ \max_{k \in \sharp_\square^i} \{(A_k \cdot (A_j - A_i \odot A_j)) - 1\}, \max_{w \in \sharp_\square^j} \{(A_w \cdot (A_i - A_j \odot A_i)) - 1\} \right\}$, with $\cdot$ being the dot product and $\odot$ the elementwise product, be the maximal number of 4-cycles based at $i \sim j$ traversing a common node.

The BFC $\kappa(i,j)$ is 0 if $\min\{d_i, d_j\} = 1$ and alternatively:

$$
\kappa(i,j) := \frac{2}{d_i} + \frac{2}{d_j} - 2 + 2 \frac{|\sharp_\Delta(i,j)|}{\max\{d_i, d_j\}} + \frac{|\sharp_\Delta(i,j)|}{\min\{d_i, d_j\}} + \frac{(\gamma_{max}(i,j))^{-1}}{\max\{d_i, d_j\}} (|\sharp_\square^i(i,j)| + |\sharp_\square^j(i,j)|).
$$
$$(12)$$

### A.2  PROOFS

For the sake of clarity and exposition, we report here both the statements and their respective proofs.

**Proposition 3.1** Define the quantity:

$$
\underline{\kappa}(i,j) = \begin{cases} 0 & \text{if } \min\{d_i, d_j\} = 1 \\ \frac{2}{d_i} + \frac{2}{d_j} - 2 & \text{otherwise} \end{cases} \quad (13)
$$

For any edge $i \sim j$ in a simple, unweighted bipartite graph $G$, we have that that $-2 < \underline{\kappa}(i,j) \leq \kappa(i,j) \leq 1$.

*Proof.* It is straightforward to see that as a direct consequence of the definition of the BFC (Def. A.2), $-2 < \underline{\kappa}(i, j) \leq \kappa(i, j)$. For the upper bound, we'll have to consider the effects of each term in the definition. First, notice that notice that the BFC on an edge $i \sim j$ of $G$ takes a simpler form compared to the more general, original definition:

$$\kappa(i, j) := \frac{2}{d_i} + \frac{2}{d_j} - 2 + \frac{(\gamma_{\max}(i, j))^{-1}}{\max\{d_i, d_j\}}(|\sharp_\square^i(i, j)| + |\sharp_\square^j(i, j)|). \tag{14}$$

This result follows from the fact that bipartite graphs have no triangles, i.e., $|\sharp_\triangle(i, j)| = 0 \forall (i, j) \in E$. Our main focus for the upper bound becomes the rightmost term. Firstly note that this term is, by definition, non-negative. We can further simplify the definitions of the 4-cycle forming neighborhoods to match the bipartite topology of $G$:

$$\sharp_\square^i(i, j) := \{n \in S_1(i) \setminus \{j\} : \exists w \in (S_1(n) \cap S_1(j)) \setminus \{i\}\}, \tag{15}$$

$$\sharp_\square^j(i, j) := \{n \in S_1(j) \setminus \{i\} : \exists w \in (S_1(n) \cap S_1(i)) \setminus \{j\}\}. \tag{16}$$

By definition of $S_1(\cdot)$ it follows immediately that the cardinality of both sets is bounded above by the node degrees:

$$|\sharp_\square^i(i, j)| \leq d_i - 1, \quad |\sharp_\square^j(i, j)| \leq d_j - 1. \tag{17}$$

Furthermore, we always have that $(\gamma_{\max}(i, j))^{-1} \leq 1$, which gives us the upper bound:

$$\kappa(i, j) \leq \frac{2}{d_i} + \frac{2}{d_j} - 2 + \frac{d_i + d_j - 2}{\max\{d_i, d_j\}}. \tag{18}$$

Let $m = \min\{d_i, d_j\}$ and $M = \max\{d_i, d_j\}$ such that w.l.o.g $M = d_i$, $m = d_j$. We can then write:

$$\kappa(i, j) \leq \frac{2}{M} + \frac{2}{m} - 2 + \frac{M + m - 2}{M} = \frac{2}{M} + \frac{2}{m} - 2 + 1 + \frac{m}{M} - \frac{2}{M} = \frac{2}{m} - 1 + \frac{m}{M}. \tag{19}$$

Note that $\frac{m}{M} \leq 1$, thus $-1 + \frac{m}{M} \leq 0$, which allows us to write $\kappa(i, j) \leq \frac{2}{m}$. Finally, ignoring the trivial case $m = \min\{d_i, d_j\} = 1$ for which $\kappa(i, j) = 0$ by definition, we have that $m \geq 2$, which allows us to arrive at the conclusion $\kappa(i, j) \leq \frac{2}{m} \leq 1$, thereby:

$$-2 < \underline{\kappa}(i, j) \leq \kappa(i, j) \leq 1. \tag{20}$$

$\square$

**Proposition 3.2** Let $i \sim j$ be an edge from the Literal-Clause Graph (LCG) representation $G$ of a random k-SAT problem with $N$ variables and $M$ clauses, with degree distributions given by Equations 5 and 3. Then $\kappa(i, j) = 0 \xrightarrow{P} 0$ as $\alpha = \frac{M}{N} \to 0$.

*Proof.* We proceed by proving the stronger statement $P(\kappa(i, j) = 0) \to 1$ as $\alpha = \frac{M}{N} \to 0$, which automatically implies $\kappa(i, j) = 0 \xrightarrow{P} 0$. A sufficient condition for proving the statement is then to show that $P(d_i = 1) \to 1$ as $\alpha = \frac{M}{N} \to 0$, given the definition of the BFC (Def. A.2). We can directly verify the probability of this event from the literal degree PMF (Eq. 5) in this limit:

$$\lim_{\alpha \to 0} P^*(d_i = 1) = \lim_{\alpha \to 0} e^{-\lambda} \frac{\lambda^0}{0!} = \lim_{\alpha \to 0} e^{-\lambda}, \tag{21}$$

where $\lambda = Mp = \frac{1}{2}\alpha k$, such that $\lim_{\alpha \to 0} e^{-\lambda} = 1$, which in turn implies that $P(d_i = 1) \to 1$ and $P(\kappa(i, j) = 0) \to 1$ as $\alpha = \frac{M}{N} \to 0$, and consequently $\kappa(i, j) = 0 \xrightarrow{P} 0$. $\square$

**Lemma 3.1** Let $i \sim j$ be an edge from the LCG representation $G$ of a random k-SAT problem with $N$ variables and $M$ clauses, with degree distributions given by Equations 5 and 3. Then $\underline{\kappa}(i, j) \xrightarrow{P} \frac{2}{k} - 2$ as $\alpha = \frac{M}{N} \to \infty$.

*Proof.* Ignoring the trivial case $k = 1$, as $\alpha \to \infty$, we have that $P^*(d_i = 1) \to 0$ (see the proof of Proposition 3.2 above), therefore we can consider the lower bound as consisting only of $\underline{\kappa}(i, j) = \frac{2}{d_i} + \frac{2}{d_j} - 2$.

Following the usual limit properties and the linearity of the expectation, we can write:

$$\lim_{\alpha \to \infty} \mathbb{E}_{(i \sim j)} \left[ \underline{\kappa}(i, j) \right] = 2 \lim_{\alpha \to \infty} \mathbb{E} \left[ \frac{1}{d_i} \right] + 2 \lim_{\alpha \to \infty} \mathbb{E} \left[ \frac{1}{d_j} \right] - 2 \tag{22}$$

By the sifting property of the delta function, we have that $\mathbb{E}[\frac{1}{d_j}] = \frac{1}{k}$ and therefore:

$$\lim_{\alpha \to \infty} \mathbb{E}_{(i \sim j)} \left[ \underline{\kappa}(i, j) \right] = 2 \lim_{\alpha \to \infty} \mathbb{E} \left[ \frac{1}{d_i} \right] + \frac{2}{k} - 2. \tag{23}$$

The only term left in order to arrive at a conclusion is $\lim_{\alpha \to \infty} \mathbb{E}[\frac{1}{d_i}]$. By definition of the expectation over a PMF we can write:

$$\lim_{\alpha \to \infty} \mathbb{E} \left[ \frac{1}{d_i} \right] = \lim_{\alpha \to \infty} \sum_{h=1}^{\infty} \frac{1}{h} e^{-\lambda} \frac{\lambda^{h-1}}{(h-1)!} = \lim_{\alpha \to \infty} \sum_{h=0}^{\infty} \frac{1}{h+1} e^{-\lambda} \frac{\lambda^h}{h!} \tag{24}$$

Letting $x \in (0, 1) \subset \mathbb{R}$, we have the equality $\frac{1}{h+1} = \int_0^1 x^h dx$, which then gives us:

$$\lim_{\alpha \to \infty} \sum_{h=0}^{\infty} \frac{1}{h+1} e^{-\lambda} \frac{\lambda^h}{h!} = \lim_{\alpha \to \infty} e^{-\lambda} \sum_{h=0}^{\infty} \left( \int_0^1 x^h dx \right) \frac{\lambda^h}{h!} = \lim_{\alpha \to \infty} e^{-\lambda} \sum_{h=0}^{\infty} \int_0^1 \frac{(x\lambda)^h}{h!} dx \tag{25}$$

Note that for $x \in (0, 1)$, we have that $\left| \frac{(x\lambda)^h}{h!} \right| < \frac{\lambda^h}{h!}$ and $\sum_{h=0}^{\infty} \frac{\lambda^h}{h!} = e^{\lambda}$, therefore by the Weierstrass M-test we have uniform convergence and thus the series can be integrated termwise, resulting in:

$$\lim_{\alpha \to \infty} e^{-\lambda} \sum_{h=0}^{\infty} \int_0^1 \frac{(x\lambda)^h}{h!} dx = \lim_{\alpha \to \infty} e^{-\lambda} \int_0^1 \sum_{h=0}^{\infty} \frac{(x\lambda)^h}{h!} dx = \lim_{\alpha \to \infty} e^{-\lambda} \int_0^1 e^{\lambda x} dx \tag{26}$$

$$= \lim_{\alpha \to \infty} e^{-\lambda} \left( \frac{e^{\lambda} - 1}{\lambda} \right) = \lim_{\alpha \to \infty} \frac{1 - e^{-\lambda}}{\lambda} = 0. \tag{27}$$

This tell us that the expected value converges as $\lim_{\alpha \to \infty} \mathbb{E}_{(i \sim j)}[\underline{\kappa}(i, j)] = -2 + \frac{2}{k}$. We now show that in the same limiting regime, the variance of the curvature lower bound vanishes, which is what we need to prove our statement. To start, notice that the variance depends entirely on the inverse literal degree term $\frac{2}{d_i}$, as the others add no variance. Therefore we start from:

$$\mathbb{V} \left[ \frac{2}{d_i} \right] = \mathbb{E} \left[ \left( \frac{2}{d_i} \right)^2 \right] - \mathbb{E} \left[ \frac{2}{d_i} \right]^2 \leq \mathbb{E} \left[ \left( \frac{2}{d_i} \right)^2 \right] = 4 \mathbb{E} \left[ \frac{1}{d_i^2} \right], \tag{28}$$

Remember that the edge-sampled literal degree PMF allows us to see the latter as a variable which $d_i = 1 + X$, $X \sim \text{Poisson}(\lambda)$. Let us consider the expectation over two disjoint event collections: $T = \{X \geq \frac{\lambda}{2}\}$ and $T^c = \{X < \frac{\lambda}{2}\}$. The idea behind this procedure is simply to analyze how the desired expectation behaves in a more "likely" scenario and in a tail scenario.

1. For $X \in T$, since $X \geq \frac{\lambda}{2}$, $1 + X \geq \frac{\lambda}{2}$, and $(1 + X)^2 \geq \frac{\lambda^2}{4}$, which finally implies $\frac{1}{(1+X)^2} \leq \frac{4}{\lambda^2}$. We can therefore write:

$$\mathbb{E} \left[ \frac{1}{d_i^2} \mathbf{1}_T \right] \leq \mathbb{E} \left[ \frac{4}{\lambda^2} \mathbf{1}_T \right] = \frac{4}{\lambda^2}, \tag{29}$$

where $\mathbf{1}_T$ is an indicator function.

2. For $X \in T^c$, we can use the crude bound $\frac{1}{(1+X)^2} \leq 1$, wich implies that $\mathbb{E}\left[\frac{1}{d_i^2}\mathbf{1}_{T^c}\right] \leq \mathbb{E}[1\ \mathbf{1}_{T^c}] = P(T^c) = P(X < \frac{\lambda}{2})$. Posed in this way, we can see that it is possible to apply a Chernoff bound on this probability, by letting $\delta = 1/2$ and remembering that $\mathbb{E}[X] = \lambda$, to obtain:

$$P(X < \frac{\lambda}{2}) \leq P(X \leq \frac{\lambda}{2}) = P(X \leq (1-\delta)\mathbb{E}[X]) \leq e^{-\delta^2 \frac{\mathbb{E}[X]}{2}} = e^{-\frac{\lambda}{8}}, \qquad (30)$$

which implies that $\mathbb{E}\left[\frac{1}{d_i^2}\mathbf{1}_{T^c}\right] \leq e^{-\frac{\lambda}{8}}$.

We can now separate the expectation, i.e., express it using partial expectations on $T$ or $T^c$:

$$\mathbb{V}\left[\frac{2}{d_i}\right] \leq 4\mathbb{E}\left[\frac{1}{d_i^2}\right] = 4\left(\mathbb{E}\left[\frac{1}{d_i^2}\mathbf{1}_T\right] + \mathbb{E}\left[\frac{1}{d_i^2}\mathbf{1}_{T^c}\right]\right) \leq 4\left(\frac{4}{\lambda^2} + e^{-\frac{\lambda}{8}}\right), \qquad (31)$$

from which it is straightforward too observe a convergence to 0 since variance is always non-negative and $\lim_{\alpha\to\infty} 4\left(\frac{4}{\lambda^2} + e^{-\frac{\lambda}{8}}\right) = 0$. We therefore have convergence at a finite expected value with the variance shrinking to 0, which implies that $\underline{\kappa}(i,j) \xrightarrow{\text{P}} \frac{2}{k} - 2$ as $\alpha \to \infty$.

$\square$

**Theorem 3.1** Let $i \sim j$ be an edge from the LCG representation $G$ of a random k-SAT problem with $N$ variables and $M$ clauses, with degree distributions given by Equations 5 and 3. Let $\sharp_\square^i(i,j) := \{n \in S_1(i) \setminus S_1(j), n \neq j : \exists w \in (S_1(n) \cap S_1(j)) \setminus S_1(i)\}$ be the neighbors of $i$ forming a 4-cycle based at $i \sim j$, *without* diagonals inside. The BFC at $i \sim j$ is bounded from above by the quantity:

$$\bar{\kappa}(i,j) := \frac{2}{d_i} + \frac{2}{d_j} - 2 + \frac{|\sharp_\square^i(i,j)| + |\sharp_\square^j(i,j)|}{d_i}. \qquad (32)$$

Furthermore, in the limit $N \to \infty$, $k$ fixed, $M = \alpha N$, and $\alpha \to \infty$, $\kappa(i,j) \xrightarrow{\text{P}} \frac{2}{k} - 2$.

*Proof.* We first prove the construction of the upper bound, which we can then use to prove the convergence via sandwiching upon the lower bound shown to converge in Lemma 3.1. We remind the reader that from the proof of Proposition 3.1 we have seen the term which constitutes the difference between $\kappa(i,j)$ and $\underline{\kappa}(i,j)$ is:

$$C(i,j) = \frac{(\gamma_{\max}(i,j))^{-1}}{\max\{d_i, d_j\}}(|\sharp_\square^i(i,j)| + |\sharp_\square^j(i,j)|). \qquad (33)$$

As another reminder, note that the definition of $\gamma_{\max}(i,j)$ can also be simplified, always as a consequence of the topology of $G$. Consider first the symmetric adjacency matrix of a bipartite graph, given by:

$$A = \begin{bmatrix} \mathbf{0} & B \\ B^T & \mathbf{0} \end{bmatrix} \in \{0,1\}^{(2N+M) \times (2N+M)}, \qquad (34)$$

where $B$ is an $2N \times M$ incidence matrix with $B_{ij} = 1$ if there if $i \sim j \in E$, and $B_{ij} = 0$ otherwise, $B^T$ is the transpose of $B$ (which ensures that $A$ is symmetric), and $\mathbf{0}$ are zero blocks of size $2N \times 2N$ and $M \times M$ corresponding to the absence of edges within the partitions. We can thus express $\gamma_{\max}(i,j)$ as:

$$\gamma_{\max}(i,j) := \max\left\{\max_{k \in \sharp_\square^i}\{A_k \cdot A_j - 1\}, \max_{w \in \sharp_\square^j}\{A_w \cdot A_i - 1\}\right\}, \qquad (35)$$

since $A_i \odot A_j = \mathbf{0}$. The operation $A_k \cdot A_j = \nu \in \mathbb{N}_0$ returns the number of common neighbors $\nu$ that nodes $k$ and $j$ have, with $k$ and $j$ being in the same partition. At this point, we can distinguish between two cases:

**Case 1: No 4-cycles.** In this case, $|\sharp_\square^i(i,j)| + |\sharp_\square^j(i,j)| = 0$ such that $C(i,j) := 0$, therefore $\kappa(i,j) = \underline{\kappa}(i,j)$ and by Lemma 3.1 we obtain $\kappa(i,j) \xrightarrow{\text{P}} -2 + \frac{2}{k}$ as $\alpha \to \infty$.

**Case 2: Presence of 4-cycles.** Firstly, note that we can again use the crude bound of the form $(\gamma_{\max}(i,j))^{-1} \leq 1$. Furthermore, $\frac{1}{\max\{d_i,d_j\}} \leq \frac{1}{d_i}$ since if $d_i \geq d_j \implies \max\{d_i,d_j\} = d_i \implies \frac{1}{\max\{d_i,d_j\}} = \frac{1}{d_i}$, while $d_i < d_j \implies \max\{d_i,d_j\} = d_j > d_i \implies \frac{1}{\max\{d_i,d_j\}} < \frac{1}{d_i}$. This gives us an upper bound on the four cycle correction term:

$$C(i,j) \leq \frac{|\sharp_\square^i(i,j)| + |\sharp_\square^j(i,j)|}{d_i} = \frac{|\sharp_\square^i(i,j)|}{d_i} + \frac{|\sharp_\square^j(i,j)|}{d_i}, \tag{36}$$

which automatically provides $\kappa(i,j) \leq \bar\kappa(i,j)$ We now inspect the final two terms separately. For the rightmost one, we have that

$$\lim_{\alpha \to \infty} \mathbb{E}\left[\frac{|\sharp_\square^j(i,j)|}{d_i}\right] \leq \lim_{\alpha \to \infty} \mathbb{E}\left[\frac{k-1}{d_i}\right] = \lim_{\alpha \to \infty} (k-1)\,\mathbb{E}\left[\frac{1}{d_i}\right] = 0, \tag{37}$$

due to the fact that $\lim_{\alpha \to \infty} \mathbb{E}[\frac{1}{d_i}] = 0$ as proved in Lemma 3.1. From the same previous result, given that $\mathbb{V}\left[\frac{k-1}{d_i}\right] \leq \mathbb{E}\left[\left(\frac{k-1}{d_i}\right)^2\right] = (k-1)^2\mathbb{E}\left[\frac{1}{d_i^2}\right]$, we have that the variance vanishes as $\alpha \to \infty$. Therefore, $\frac{|\sharp_\square^j(i,j)|}{d_i} \xrightarrow{P} 0$ as $\alpha \to \infty$.

We are now left with the final term, for which we need to make some additional considerations. In particular, we'll closely inspect the set $\sharp_\square^i(i,j)$ and provide a combinatorial argument for its behavior. Let us start by asking what is the probability that *a random clause $n$ containing $i$ shares at least one other literal with $j$*. For a given value of $N$ we will denote this by $P(\text{cycle overlap}) = \omega_N$. For the edge $i \sim j$ with $d_i = h$, there are $h-1$ candidate clauses that can belong in $\sharp_\square^i(i,j)$, which means $n$ can be any of these nodes. Then, the inclusion of $n$ in the set depends on whether there exists at least a literal which $n$ shares with $j$ that is not $i$. Equivalently, we want to *understand the probability of any of $n$'s other $k-1$ literals being among the remaining $k-1$ literals of $j$*.

Let us flip the problem and ask ourselves about the probability of no cycle overlap first: this implies that clause $n$ has to choose $k-1$ literals out of $2N-k$, given that out of the $2N$ available literals, the $k$ present in clause $j$ must be ignored. On the other hand, given that each clause chooses $k$ distinct literals uniformly, there are $k-1$ literals, ignoring $i$, that can be a part of $n$ out of $2N-1$ total. Using the aforementioned logic, the probability of no overlap for any candidate clause can be written as:

$$P(\text{no cycle overlap}) = \frac{\binom{2N-k}{k-1}}{\binom{2N-1}{k-1}} = \frac{\frac{(2N-k)(2N-k-1)\ldots(2N-k-(k-2))}{(k-1)!}}{\frac{(2N-1)(2N-2)\ldots(2N-1-(k-2))}{(k-1)!}} \tag{38}$$

$$= \prod_{z=0}^{k-2} \frac{2N-k-z}{2N-1-z} = \prod_{z=0}^{k-2} 1 - \frac{k-1}{2N-1-z} = 1 - \omega_N. \tag{39}$$

At this point, we can make a scaling argument by using the sparsity that the assumed asymptotic regime offers, given that we consider the graphs to have $k$ fixed while $N \to \infty$, such that $N \gg k$. This allows us to consider $\frac{k-1}{2N-1-z} = O(\frac{1}{N})$ and therefore $\omega_N \to 0$.

Remember that for the edge $i \sim j$ with $d_i = h$, there are $h-1$ candidate clauses that can belong in $\sharp_\square^i(i,j)$, therefore by conditioning and a simple counting argument we can write:

$$\mathbb{E}\left[|\sharp_\square^i(i,j)| \mid d_i = h\right] = (h-1)\omega_N \implies \mathbb{E}\left[\frac{|\sharp_\square^i(i,j)|}{d_i} \mid d_i = h\right] = \frac{(h-1)}{h}\omega_N \leq \omega_N. \tag{40}$$

Note that this bound is constant in $h$, therefore by the law of total expectation we have that:

$$\mathbb{E}\left[\frac{|\sharp_\square^i(i,j)|}{d_i}\right] \leq \omega_N \to 0. \tag{41}$$

Given that the quantity $\frac{|\sharp_\square^i(i,j)|}{d_i} \geq 0$ we can apply Markov's inequality to obtain, for any $\epsilon > 0$:

$$P(\frac{|\sharp_\square^i(i,j)|}{d_i} \geq \epsilon) \leq \frac{\mathbb{E}\left[\frac{|\sharp_\square^i(i,j)|}{d_i}\right]}{\epsilon} \leq \frac{\omega_N}{\epsilon} \to 0. \tag{42}$$

This result is particularly insightful, because it shows that the behavior of the term $\frac{|\sharp_\square^i(i,j)|}{d_i}$ does not depend explicitly on $\alpha$, but it is in fact the appropriate scaling regime (i.e., the thermodynamic limit with k fixed) that induces sparsity and therefore diminishes the overlap probability. Nevertheless, given we work in this regime, we can overload notation somewhat and write that $\frac{|\sharp_\square^i(i,j)|}{d_i} \xrightarrow{\text{P}} 0$ as $\alpha \to \infty$.

Finally, because $\bar{\kappa}(i,j) = \frac{2}{d_i} + \frac{2}{d_j} - 2 + \frac{|\sharp_\square^i(i,j)|}{d_i} + \frac{|\sharp_\square^j(i,j)|}{d_i} \geq \kappa(i,j) \geq \underline{\kappa}(i,j)$, and both $\bar{\kappa}(i,j) \xrightarrow{\text{P}} \frac{2}{k} - 2$ and $\underline{\kappa}(i,j) \xrightarrow{\text{P}} \frac{2}{k} - 2$ as $\alpha \to \infty$, we can conclude that $\kappa(i,j) \xrightarrow{\text{P}} \frac{2}{k} - 2$ as $\alpha \to \infty$.

$\square$

### A.3 DETAILS ON TEST-TIME REWIRING

Graph rewiring is the process of modifying a graph's connectivity by adding, removing, or re-weighting edges, typically to optimize information flow. In our case, we make use of the rewiring procedure presented by Topping et al. (2022), which consists of a discrete and stochastic Ricci flow, guided by the BFC. At each iteration, the edge with the most negative curvature (i.e., the structurally most "strained" connection) is identified. Candidate rewiring edges are then generated between the neighborhoods of the two endpoints of this edge. A new edge is stochastically selected either at random (with probability $p$) or by maximizing the curvature improvement obtained from its addition. The selected edge is added to the graph and the corresponding curvature values are updated. By repeating this procedure for a fixed number of iterations, the algorithm progressively increases the average curvature of the graph, yielding a rewired version that contains information bottlenecks that are weaker compared to the input. Algorithm 1 contains the pseudocode for our rewiring algorithm. We would like to stress here that this modifies the constraints of the Boolean Satisfiability Problem (SAT) problem under consideration, but the goal of this procedure is to show that "flatter" problems will in fact be easier to solver for Graph Neural Network (GNN)-based solvers even without retraining.

---

**Algorithm 1:** Balanced Forman Curvature Stochastic Rewiring

**Input:** Graph $G = (V, E)$ with edge BFC values $\kappa(i,j), \; \forall (i,j) \in E$, probability value $p \in [0,1]$, number of iterations $N \in \mathbb{N}$

**Output:** Rewired graph $G'$ with updated BFC values

**for** $t \leftarrow 1$ **to** $N$ **do**

    Select edge $(i,j)$ with most negative curvature $\kappa(i,j)$;

    From the neighbors $S_1(i)$ and $S_1(j)$ form candidate edge set

$$C = \{(k,l) : k \in S_1(i), l \in S_1(j), (k,l) \notin E\}$$

    **if** $C = \emptyset$ **then**

      $\llcorner$ continue

    With probability $p$, choose a random edge $(k,l) \in C$;

    Otherwise, for each $(k,l) \in C$:

        1. Compute updated curvature $\kappa'(i,j)$ after adding $(k,l)$.

        2. Evaluate improvement score $\Delta_{kl} = -(\kappa(i,j) - \kappa'(i,j))$.

    Select $(k,l)$ with maximum $\Delta_{kl}$;

    Add edge $(k,l)$ to $G$ and update neighborhood sets;

    Update curvatures $\kappa(i,j)$ and $\kappa(k,l)$ accordingly;

Finalize bipartite structure of literals and clauses $(L, C)$ ensuring no intra-partition edges and set $G' = G$ at time $t = N$;

**return** $G'$

---

### A.4 SOME INITIAL IDEAS FOR CURVATURE-AWARE SOLVERS

As discussed during the conclusion, we showed that the graph BFC displays concentration to particular constants with high probability. This result makes it legitimate to conjecture that using a curvature-

Table 3: Accuracy of producing satisfying assignments on SAT benchmark datasets (Li et al., 2024) of two different GNN-based solvers with different message-passing schemes: 1) Vanilla uses the typical message-passing operation; 2) Curvature Gate learns a gating function for each edge based on its curvature value (Ye et al., 2019); 3) Online LCP extends the work of Fesser & Weber (2024) and concatenates the local curvature statistics around each nodes as features during each recurrent step; 4) Both uses Curvature Gate and Online LCP. The reported results represent the average and standard deviation over 5 different runs, with the **best results** (for each model) in **boldface**.

| Model | Variation | Datasets | | | |
|-------|-----------|----------|----------|----------|----------|
|       |           | 3-SAT | 4-SAT | SR | CA |
| GCN | Vanilla | $0.510 \pm 0.012$ | $\mathbf{0.180} \pm 0.048$ | $\mathbf{0.470} \pm 0.031$ | $0.650 \pm 0.016$ |
|  | + Curvature Gate | $\mathbf{0.514} \pm 0.018$ | $0.154 \pm 0.017$ | $0.422 \pm 0.013$ | $\mathbf{0.664} \pm 0.042$ |
|  | + Online LCP | $0.510 \pm 0.014$ | $0.170 \pm 0.010$ | $0.422 \pm 0.016$ | $0.662 \pm 0.016$ |
|  | + Both | $0.500 \pm 0.012$ | $0.176 \pm 0.031$ | $0.416 \pm 0.027$ | $0.654 \pm 0.027$ |
| NeuroSAT | Vanilla | $0.690 \pm 0.022$ | $0.436 \pm 0.032$ | $0.734 \pm 0.017$ | $0.746 \pm 0.018$ |
|  | + Curvature Gate | $0.682 \pm 0.030$ | $0.436 \pm 0.015$ | $\mathbf{0.742} \pm 0.027$ | $\mathbf{0.758} \pm 0.016$ |
|  | + Online LCP | $\mathbf{0.692} \pm 0.020$ | $0.416 \pm 0.046$ | $0.724 \pm 0.022$ | $0.726 \pm 0.059$ |
|  | + Both | $0.664 \pm 0.018$ | $\mathbf{0.438} \pm 0.028$ | $\mathbf{0.742} \pm 0.025$ | $\mathbf{0.758} \pm 0.020$ |

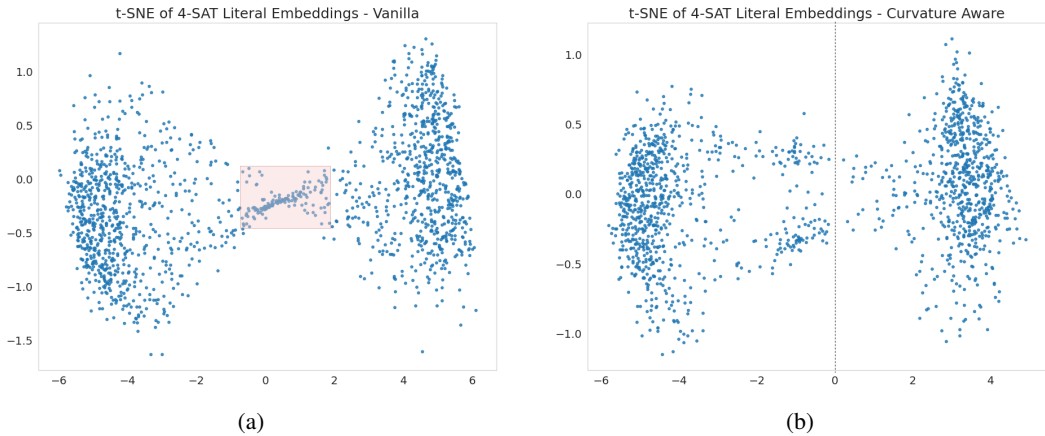

(a)        (b)

Figure 2: Low dimensional visualization of the literal embeddings produced by NeuroSAT on random 4-SAT with (a) vanilla message passing and (b) Curvature Gate. Even though there is no major change in performance, the learned representations can be linearly separated into truth value assignments in the curvature-aware case, indicating promise for the inclusion of these principles in future GNN-based solver design.

aware message passing procedure naively would not lead to stable performance benefits. In this section, we provide some starting points and experiments to confirm this conjecture. The goal of our implementations was to maintain efficiency and rely on straightforward ideas that could lead to performance improvements. For these purposes, we introduce two simple curvature-aware variants of message passing. The first is an adaptation of Ye et al. (2019), where the curvature of an edge is used to learn a gating mechanism that modulates message contributions. The second is a simple recurrent extension of Fesser & Weber (2024), where the statistics of the curvature around each node are used as additional features at each recurrent step. Our empirical findings reveal that naively injecting curvature into GNN-based solvers sometimes leads to improved performance, but it does not always provide clear advantages, as seen in Table 3. While we believe that both these implementations provide interesting starting points for future work and research, another direction is understanding how to effectively alter the curvature in data space, in a way that is compatible with SAT problems.

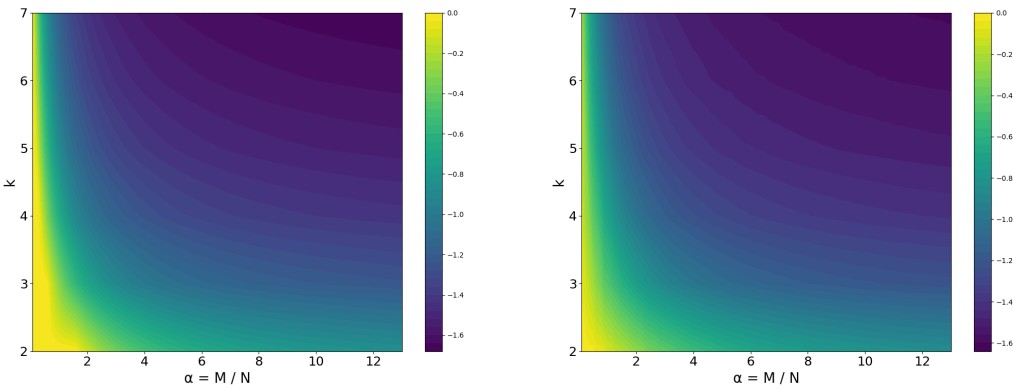

(a) Average graph BFC lower bound $\underline{\kappa}(i,j)$ as a function of $k$ and $\alpha$.

(b) Average graph BFC $\kappa(i,j)$ as a function of $k$ and $\alpha$.

Figure 3: Average graph Ricci Curvature lower bound (a) and Balanced Forman Curvature (b) as a function of $\alpha$ and $k$. Both quantities behave very similarly, both in terms of the smooth transition from flat to negative curvature and their magnitude, especially as $\alpha \to 0$ and $\alpha \to \infty$, in line with the developed theory. Best viewed in color.

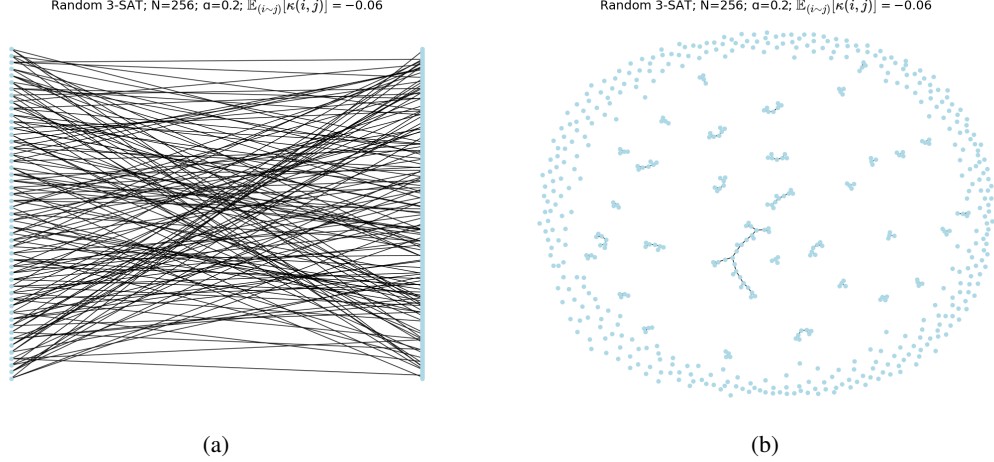

(a)                                         (b)

Figure 4: Visualization of an easy (in terms of clause density $\alpha$) random 3-SAT problem with 256 variables in (a) bipartite and (b) circular layouts. There is a very small number of long-range interactions, as can clearly be seen in (b). Furthermore, for such very small $\alpha$, the average BFC approaches 0, in line with the developed theory.

## A.5  ADDITIONAL PLOTS

In this section we provide additional and miscellaneous plot that help with visualizing various concepts explained in the main manuscript, such as the relationship between curvature and hardness, as well as visualizations of easy and hard problems. More details can be found in Figures 3, 4, 5 and 6.

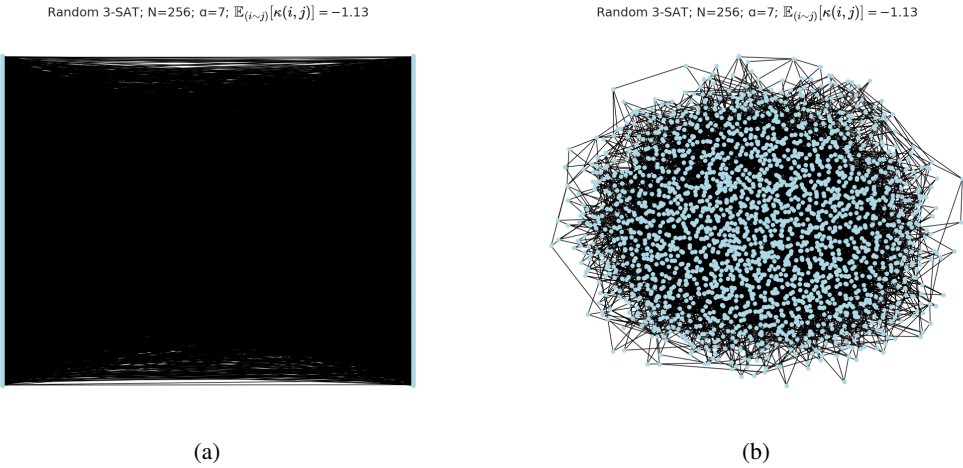

(a)                                                                    (b)

Figure 5: Visualization of a hard (in terms of clause density $\alpha$) random 3-SAT problem with 256 variables in (a) bipartite and (b) circular layouts. There is a very large number of long-range interactions, as can clearly be seen in (b). Furthermore, for such large $\alpha$, the average BFC is strongly negative, in line with the developed theory.

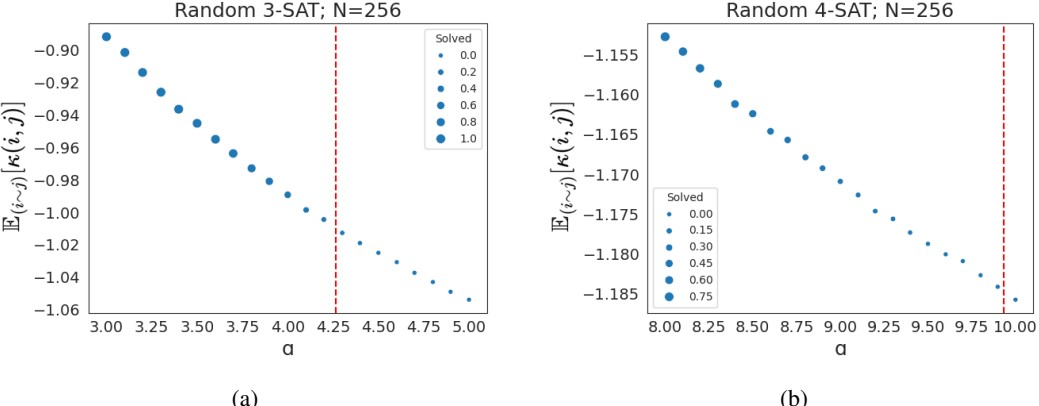

(a)                                                                    (b)

Figure 6: Average BFC as a function of $\alpha$ for random 3 and 4 -SAT problems with $N = 256$. The size of the blobs is used as a representation for the average solvability of the problems at a given $\alpha$ by the NeuroSAT model (Selsam et al., 2019). The vertical red line represents the analytical SAT-UNSAT critical threshold $\alpha_c$ (Mertens et al., 2006). The average BFC drops monotonically with $\alpha$ in both cases. For 3-SAT (a), this drop appears to be almost linear, with a substantial amount of variance, as also depicted in Figure 1. 4-SAT on the other hand contains problems where the curvature is substantially more negative and concentrated, which when coupled with the larger number of constraints leads to oversquashing, as discussed in our theory. This can be clearly seen by the fast performance drop-off in (b), which happens much earlier than $\alpha_c$, indicating the additional hardness phase related to representation learning discussed in the main paper.

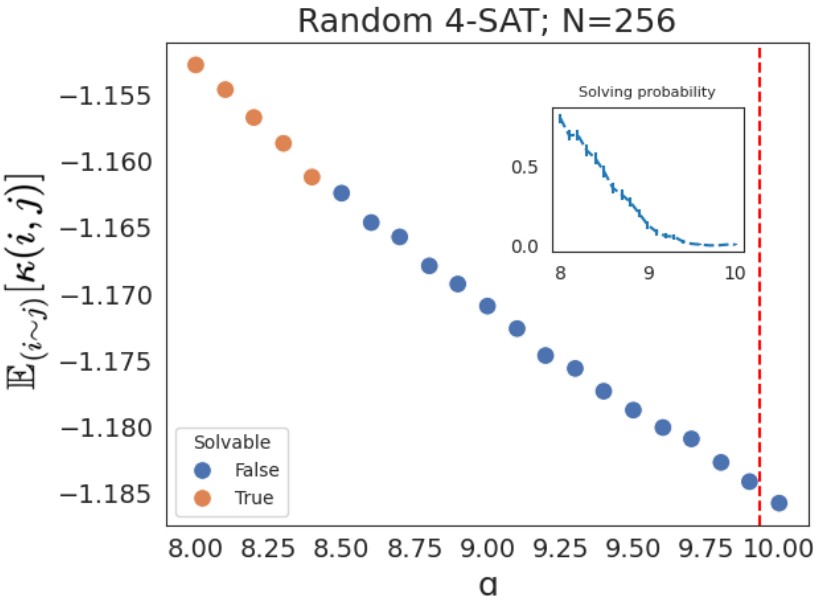

Figure 7: Analogue of Figure 1a for 4-SAT. Average BFC as a function of $\alpha$ for random 4-SAT problems with $N = 256$. The color is used as a representation for the average solvability of the problems at a given $\alpha$ by the NeuroSAT model, with a group labeled as solvable if 50% or more of the problems get a satisfying assignment. The vertical red line represents the analytical SAT-UNSAT critical threshold $\alpha_c \approx 9.931$ (Mertens et al., 2006). The average BFC is negative and drops with $\alpha$, but for a small amount. The plot in the top-right corner shows the model's solution probability curve as a function of $\alpha$, where it is possible to notice that the GNN-based solver has severe limitations on these problems. Differently from 3-SAT (Figure 1a), the average curvature is negative and concentrated even for problems that would be considered simple in terms of $\alpha$ ($\alpha \to 8$). This is due to the larger value of $k$, the higher number of long-range interactions, and the connection of both factors with oversquashing, as discussed throughout the paper.

