# OpenReview forum: "A Geometric Perspective on the Difficulties of Learning GNN-based SAT Solvers"
_ICLR.cc/2026/Workshop/GRaM — ICLR 2026 Workshop GRaM Poster_

### Official Review · Reviewer_aKm1 · 2026-02-11
**Review of the paper**

**Rating:** 6
**Confidence:** 3

**Review:**

Summary:
- This paper studies why GNN-based SAT solvers degrade on harder instances through a geometric lens.
- Empirically, the authors show: (1) a phase-transition-like relationship between Ricci curvature statistics and solve probability, (2) that test-time curvature-reducing rewiring improves accuracy without retraining, (3) and that two curvature-based heuristics predict generalization error better than average across several SAT benchmarks.

Strengths:
- The paper is generally well written and the argumentation line is easy to follow.
- Derives probabilistic characterizations of BFC in the regimes $\alpha\to0$ and $\alpha\to\infty$.
- The employed approach is novel in order to understand SAT solvers (i.e., through connecting discrete curvature theory and SAT structure).
- The paper offers theoretically grounded perspective on why GNN SAT solvers struggle with harder instances.


Weaknesses:
- The asymptotic limit where BFC $\to 2/k − 2$ assumes the 4-cycle term vanishes or becomes negligible; this is plausible but not fully justified given that 4-cycles proliferate in dense bipartite graphs
- The data model assumes independent random clause construction - how realistic it is?
- The paper could better position itself relative to recent critical assessments of curvature-based rewiring.

Misc:
- There are some types (e.g., line 30: "wether" should be "whether"), and missing spaces between citations (e.g., line 33: "..problem(Cook, 1971)..")
- Sometimes wording is slightly exaggerated for a scientic publication (e.g., line 217 ".. extremely interesting...", line 404: "...extremely fascinating...")
- Figure 1: the legend is too small, additionally Fig. 1(a) is rather sparse

**Pmlr Suitability:**

Yes

---

### Official Review · Reviewer_UFzu · 2026-02-23
**Geometric diagnosis of GNN-based SAT solvers that is relevant and insightful but limited in scope**

**Rating:** 6
**Confidence:** 3

**Review:**

Relevance: Highly relevant to GRaM because it uses graph Ricci curvature to diagnose a concrete limitation of message-passing GNNs on SAT graphs. It also positions curvature as a dataset/instance descriptor linked to generalization and difficulty.

Novelty: The novelty is the probabilistic characterization of BFC behavior in random k-SAT bipartite graphs and its use to motivate oversquashing-driven failure modes for GNN solvers. The curvature-moment heuristics for predicting generalization error and the test-time curvature manipulation are also fresh angles. The curvature-oversquashing link is borrowed from prior theory.

Soundness: The theoretical analysis is coherent and mathematically grounded within the random k-SAT model, though its generalization beyond this setting remains an assumption. Empirical evidence consistently supports the curvature–performance relationship. The rewiring results further suggest that alleviating bottlenecks improves solver behavior without retraining, albeit at the cost of altering the instance structure.

Clarity: The paper is well structured.

Overall assessment: This is a good GRaM submission. The main risk is overextending the theoretical conclusions beyond the random model and interpreting empirical correlations as strict causal mechanisms.

**Pmlr Suitability:**

Yes

---

### Official Review · Reviewer_eDMA · 2026-02-24
**Recommendation: Accept**

**Rating:** 9
**Confidence:** 2

**Review:**

This is a high-quality, original and clear paper that is a pleasure to read.

The authors demonstrate a specific failure case of using GNNs for SAT solvers -- the negative curvature of the associated bipartite graph. The authors prove several theorems and show several clear and convincing empirical studies, including real-world datasets.
The background section is very well written.
Despite the highly technical content, the paper is understandable to non-experts.
The scope is limited but the exposition is clear and precise.

The cons are:
- only the uniformly-at-random model is analyzed
- no solution is provided for the identified problem

However, I believe that both can be left for future work, as the paper is already quite dense.

**Pmlr Suitability:**

Yes

---

### Meta-Review · Area_Chair_LSiZ · 2026-02-26

**Decision:**

Accept

**Metareview:**

The paper demonstrates a specific failure case of using GNNs for SAT solvers. Reviewers appreciates the novelty of the approach, sound theoretical analysis, and high writing quality. However, the setting of the analysis is limited and assumptions are sometimes not clearly justified. Overall, I recommend acceptance and encourage the authors to incorporate reviewers’ feedbacks in the final version.

**Relevance To Proceedings:**

Yes — suitable for PMLR (long paper)

**Relevance To Workshop:**

Yes — suitable for GRaM

---

### Decision · Program_Chairs · 2026-03-02

Accept (Poster)